Meloxicam ameliorates the cartilage and subchondral bone deterioration in monoiodoacetate-induced rat osteoarthritis

Nagy Előd elod.nagy@umftgm.ro 1
Vajda Enikő 2
Vari Camil 3
Sipka Sándor 4
Fárr Ana-Maria 5
Horváth Emőke 6
1 Department of Biochemistry and Environmental Chemistry, University of Medicine and Pharmacy , Targu-Mures , Romania
2 Department of Drug Analysis, University of Medicine and Pharmacy , Targu-Mures , Romania
3 Department of Pharmacology, University of Medicine and Pharmacy of Targu Mures , Targu-Mures , Romania
4 Division of Clinical Immunology, Department of Internal Medicine, University of Debrecen , Hungary
5 Department of Pathophysiology, University of Medicine and Pharmacy , Targu-Mures , Romania
6 Department of Pathology, University of Medicine and Pharmacy , Targu-Mures , Romania
Rocha Joao
Electronic publication date: 2017 Apr 12
Publication date: 2017
Volume: 5
Electronic Location ID: e3185
Received 2016 Nov 4; Accepted 2017 Mar 14
Copyright: ©2017 Nagy et al.
Copyright year: 2017
Copyright holder: Nagy et al.
License: This is an open access article distributed under the terms of the Creative Commons Attribution License, which permits unrestricted use, distribution, reproduction and adaptation in any medium and for any purpose provided that it is properly attributed. For attribution, the original author(s), title, publication source (PeerJ) and either DOI or URL of the article must be cited.
License URL: https://creativecommons.org/licenses/by/4.0/

Keywords: Meloxicam, Subchondral bone, Osteoarthritis, Cox-2, Inflammation, Mono-iodoacetate, OARSI histopathology initiative

Funding: Sectorial Operational Programme Human Resources Development (SOPHRD) European Social Fund Romanian Government POSDR 80641 Studium-Prospero Foundation 6/30.10.2013 0345/26.02.2016 All the funding of this work was supported by the Sectorial Operational Programme Human Resources Development (SOPHRD), financed by the European Social Fund and by the Romanian Government under the contract number POSDR 80641, and research contract no. 6/30.10.2013 and 0345/26.02.2016 with the Studium-Prospero Foundation. There was no additional external funding received for this study. The funders had no role in study design, data collection and analysis, decision to publish, or preparation of the manuscript.

==============================
Objective

This study aimed to quantify the cartilage- and subchondral bone-related effects of low-dose and high-dose meloxicam treatment in the late phase of mono-iodoacetate-induced osteoarthritis of the stifle.

Methods

Thirty-four male Wistar rats received intra-articular injection of mono-iodoacetate to trigger osteoarthritis; 10 control animals (Grp Co) received saline. The mono-iodoacetate-injected rats were assigned to three groups and treated from week 4 to the end of week 7 with placebo (Grp P, n = 11), low-dose (GrpM Lo, 0.2 mg/kg, n = 12) or high-dose (GrpM Hi, 1 mg/kg, n = 11) meloxicam. After a period of 4 additional weeks (end of week 11) the animals were sacrificed, and the stifle joints were examined histologically and immunohistochemically for cyclooxygenase 2, in conformity with recommendations of the Osteoarthritis Research Society International. Serum cytokines IL-6, TNFα and IL-10 were measured at the end of weeks 3, 7, and 11.

Results

Compared with saline-treated controls, animals treated with mono-iodoacetate developed various degrees of osteoarthritis. The cartilage degeneration score and the total cartilage degeneration width were significantly lower in both the low-dose (p = 0.012 and p = 0.014) and high-dose (p = 0.003 and p = 0.006) meloxicam-treated groups than in the placebo group. In the subchondral bone, only high-dose meloxicam exerted a significant protective effect (p = 0.011). Low-grade Cox-2 expression observed in placebo-treated animals was abolished in both meloxicam groups. Increase with borderline significance of TNFα in GrpP from week 3 to week 7 (p = 0.049) and reduction of IL-6 in GrpM Lo from week 3 to week 11 (p = 0.044) were observed.

Conclusion

In this rat model of osteoarthritis, both low-dose and high-dose meloxicam had a chondroprotective effect, and the high dose also protected against subchondral bone lesions. The results suggest a superior protection of the high-dose meloxicam arresting the low-grade inflammatory pathway accompanied by chronic cartilage deterioration.

Introduction

Osteoarthritis (OA) is a complex chronic disorder characterized by loss and metabolic changes of the cartilage matrix along with low-grade inflammation and alterations of the subchondral bone. The degradative pathways were initially thought to be cartilage-driven, but newer evidence has documented that they are substantially influenced by inflammatory mediators released from the subchondral bone (Westacott et al., 1997; Berenbaum, 2013; Yu et al., 2016). An up-regulated signalling cross-talk between the osteoarthritic subchondral bone and cartilage has been revealed (Yuan et al., 2014). In experimental osteoarthritis of the stifle, intra-articular injection of mono-iodoacetate (MIA), a glyceraldehide-3-phosphate dehydrogenase inhibitor, halts the glycolysis, provoking chondrocyte death (especially in the central tibial zone), loss of proteoglycans, fibrillation, and formation of cysts and large osteophytes (Bendele, 2001). The chronologic development of histological lesions in MIA-induced osteoarthritis has been described in detail (Guzman et al., 2003). The predominant early symptoms of MIA toxicity are shrinking, degeneration and death of chondrocytes, with synovial edema and a moderate mononuclear cell infiltrate (Guzman et al., 2003). After a week, clustered osteoblasts and increased osteoclast activation can be observed, indicating increased bone remodelling. Cartilage fragmentation and erosion in tight anatomic relationship with collapse and fragmentation of the underlying bony trabeculae are present (Guzman et al., 2003). These histological changes occur on the background of variations in biphasic proteoglycan synthesis (Dumond et al., 2004).

After MIA injection, cyclooxygenase-2 (Cox-2), together with matrix metalloproteinase-2 activity, is quickly up-regulated. Some important pro-inflammatory genes, such as interleukin-1β (IL-1β), and inducible nitric oxide synthase (iNOS), also had increased activity in a MIA-model, in parallel with the down-regulation of proteoglycan synthesis in the tibial plateau and condyle (Dumond et al., 2004). Experimental, but also human osteoarthritis are characterized by a self-perpetuating, low-grade inflammation affecting both the synovial membrane and the cartilage. Regulatory molecules, such as IL-1β, leptin, TNFα, receptor for advanced glycation end-products (RAGE), and iNOS, confer a destruction-prone activated phenotype to chondrocytes (Musumeci et al., 2015).

Orally administered Cox-2 selective non-steroidal anti-inflammatory agents may decrease persistent low-grade inflammation. These drugs are recommended in the recent Osteoarthritis Research Society International (OARSI) therapeutic guidelines for the treatment of knee-only osteoarthritis in patients who have no co-morbidities or multi-joint osteoarthritis (McAlindon et al., 2015).

However, there is a need for more group-specific therapies, as orally administered meloxicam may be associated with serious dose-related adverse effects and lower-dose pharmaceutical formulations are under evaluation in phase 3 clinical studies (Altman et al., 2015). As an alternative, experimental approaches with intra-articular repeated high doses of the drug proved to be successful (Wen et al., 2013).

The grading system elaborated by Pritzker et al. (2006) proved to offer simplicity, scalability, extendability and comparability to the histopathology evaluation (Custers et al., 2007). In experimental conditions, detailed recommendations for the histopathological examination of joint lesions of the rat were elaborated through the OARSI histopathology initiative (Gerwin et al., 2010) in order to allow standardization and a better comparison of various results.

Since the histological changes in MIA-triggered osteoarthritis are progressive and the degenerative process in the subchondral bone peaks late, from day 42 to 56 (Guzman et al., 2003; Miyamoto et al., 2016), we considered that this period corresponds to the late, chronic phase of osteoarthritis. Further, this time period corresponds with the estimation that 11 weeks of rat age equals several years in humans (Sengupta, 2013). We have hypothesized that low-grade inflammation induced by MIA persists in the subchondral bone and bone marrow in the late-phase reaction of osteoarthritis, thus sustaining cartilage destruction. We proposed that meloxicam can ameliorate not only the cartilage deterioration, but also the lesions provoked on subchondral bone. Our goal was to quantify the dose-related effects of meloxicam treatment in the subchondral bone-involving, late phase of mono-iodoacetate-induced osteoarthritis, and to follow the levels of serum IL-6, IL-10 and TNFα, as markers of systemic inflammation.

In order to elucidate whether a daily single dose of meloxicam will maintain circulating levels, and to establish the time to reach the steady-state plasma concentration, we performed a pharmacokinetic pre-study of meloxicam.

Methods

Experimental design

The experimental work was designed and performed in conformity with the ARRIVE guidelines (Kilkenny et al., 2010). The major phases, grouping, timing of interventions, and checkpoints of the experiment are summarized in a flow-chart below (Fig. 1).

Figure 1 Flowchart representation of the experimental design and timeline.

OA, osteoarthritis; K–L grade, Kellgren–Lawrence grade; MXC LD, meloxicam, low-dose; MXC HD, meloxicam, high-dose.

Animals

The study obtained the approval of the Ethics Committee of the University of Medicine and Pharmacy Targu-Mures (no.132/23.12.2013). The sample size was calculated by simulations assuming different means and slightly different variability of the two major histological scores and IL-6 levels between the groups, setting α = 0.05 and 1-β 0.8. Initially, 46 healthy adult male Wistar rats (age: six months, body weight: 170–230 g), obtained from the University Biobase, were entered into the study. The animals were kept under controlled conditions (temperature 22.5 ± 2 °C, humidity 55 ± 5%) on a 12-h light-dark cycle, 5–6 animals per cage (solid bottom), and fed with standard rat chow and water ad libitum. All procedures and experimental protocols used in this study were in compliance with the 2010/63/EU Directive of the European Parliament and Council accepted at September 22, 2010, for the protection of animals used for scientific purposes.

The osteoarthritis model

Forty-six animals were assigned to one of four groups by a simple randomization. The first group (GrpCo), consisting of 10 animals, was assigned to control conditions; animals in this group received only 50 µl of intra-articular saline. The other 36 animals received sodium mono-iodoacetate (MIA; Biochemica, Applichem Gmbh, Germany), 4 mg dissolved in 50 µl of physiological saline, injected into the joint cavity through the infrapatellar ligament of the left stifle, with a 26-gauge, 0.5 inch needle. The intra-articular injection was performed under anaesthesia with ketamine 10% (80 mg/kg) plus xylazine 1% (5 mg/kg). After MIA injection, animals were carefully inspected daily by trained personnel blinded to treatment data to assess stifle-joint swelling and dysfunction.

At the end of the first and the third weeks after the injection, the presence of osteoarthritis in each animal was confirmed by stifle radiography performed by a veterinarian using Univet LX 120 veterinary X-ray equipment. Radiological grading of the stifle joint lesions was done according to the K–L grading system (Kellgren & Lawrence, 1957). Two animals with low grades and no clear radiographic signs of osteoarthritis were excluded from the study.

After three weeks, the remaining 34 animals were treated with an anti-inflammatory protocol as follows:

GrpP (n = 11) received placebo solution (polyethylene glycol 400/water solution 50:50) each day; GrpM Lo (n = 12) received low-dose meloxicam (0.2 mg/kg/daily); GrpM Hi (n = 11) received high-dose meloxicam (1 mg/kg/daily). The high-dose of meloxicam (1 mg/kg) has been chosen considering the human equivalent dose (HED = 0.162 × 1 mg/kg) (Nair & Jacob, 2016), while the low-dose was determined taking into consideration toxicological data indicating 0.2 mg/kg as the “No Effect Level” on the gastrointestinal system and the kidney (Summary Report, Committee for Veterinary Medicinal Products, The European Agency for the Evaluation of Medicinal Products, Meloxicam, June 1997). The drug was dissolved in a polyethylene glycol 400/water solution (50:50) vehicle and administered daily by gavage, in a single dose between 9 and 10 a.m. for period of four weeks.

Pharmacokinetic study of meloxicam

Six Wistar rats aged six months, with a weight of 363 ± 12 g, were kept at 24 ± 2 °C and 55 ± 5‰ humidity. A 1 mg/kg dose of meloxicam was administered by oral gavage. Blood samples with a maximum volume of 150 μL were drawn from the caudal vein at 1,2,6,8,12,24,36, and 48 h after the drug delivery in heparinized tubes. Plasma was separated by centrifugation at 3,500 rpm, 10 min.

For the quantification of meloxicam, a self-developed method was applied: plasma samples were diluted 1:10 with blank pooled plasma, internal standard was added, and deproteinizing solution was applied. The diluted sample/internal standard/acetonitril/ HClO4 mixture was prepared in a proportion of 16:4:10:3, then vortexed for 15 s, allowed to stand for 5 min, and centrifuged 10 min at 10,000 rpm. The supernate was analyzed on a Waters Symmetry C-8 chromatography column, 4.6 × 150 mm, 5 μm with a Merck-Hitachi-LaChema instrument. The elution buffer was 1% aqueous solution of acetic acid:acetonitril (60:40), yield 1.7 mL/min; detection was performed by diode array detector (DAD) at 355 nm.

The pharmacokinetic parameters were determined by use of the software Kinetica 5.1 SP1 (Thermo Fisher Scientific Inc.). The AUC was calculated with a mixed log-linear method, and all of the pharmacokinetic parameters were calculated by applying a non-compartmental model.

Blood sampling

Baseline blood sampling was performed three weeks after the MIA injection, before start of meloxicam treatment. A second blood sampling took place at the end of meloxicam treatment, and the last one after four weeks of rest. Each time, 150 μL of venous blood was drawn from the tail vein and centrifuged at 3,500 rpm for 10 min. Serum aliquots were stored at −20 °C until the determination of IL-6, Il-10 and TNFα.

Serum cytokine measurement

The thawed serum aliquots were used for the measurement of IL-6, TNFα, and IL-10, which was performed on a Luminex 200 platform, applying Fluorokine Map Rat IL-6 (LUR506), IL-10 (LUR522), and TNFα/TNFSF2 (LUR510) kits (R&D Systems, Minneapolis, MN, USA).

Histological analysis of the joints

At the end-point, four weeks after meloxicam treatment was terminated, all animals were sacrificed by excessive halothane inhalation, realized with an anaesthetic vaporizer (halothane saturation 6%, O2 1.5 L/min), followed by cervical dislocation. After absence of vital functions had been determined by the veterinarian, osteoarticular samples were obtained from each animal from the injected- and the contralateral side (limb). After soft tissue was removed, the remaining material was fixed with formalin (4% neutral-solution) for three days and decalcified in Richard-Allan Scientific Decalcifying solution (Thermo Scientific, Kalamazoo, MI, USA) for two days. The specimens were processed by the frontal sectioning method followed by paraffin embedding (Gerwin et al., 2010). From each sample, serial 4-µm sections were cut in about 300-μm steps, followed by haematoxylin-eosin (HE) and periodic-acid-Schiff (PAS) staining. Microscopic examination was performed by two independent investigators, blinded for the group classification, with synchronous imaging on a trinocular microscope. The contralateral limb in every case was investigated as a negative control, with a single section per each healthy joint.

On each section, four scores proposed by the OARSI histopathology initiative were determined: the cartilage degeneration score (CDS), the total cartilage degeneration width (TCDW), the calcified cartilage and subchondral bone damage score (SBD), and the synovial reaction (SR) (Gerwin et al., 2010). The average values obtained from the two investigators were used for statistical analysis. Inter-observer Spearman correlation coefficients for these parameters were the following: r(grade) = 0.97, r(stage) = 0.90, r(CDS) = 0.96, r(TCDW) = 0.98, r(CCSBD) = 0.94, r(SR) = 1.

Cox-2 immunohistochemistry

Cox-2 was stained on a single section from each animal by applying a monoclonal anti-Cox2 antibody (SP21; Thermo Fisher Scientific) in a dilution of 1:100, with high pH retrieval solution, in conformity with the heat-induced epitope retrieval method, followed by 60-min incubation at room temperature. The reaction product was developed with the reagent Envision-FLEX High pH (Dako, Glostrup, Denmark) and made visible by reaction with diamino-benzidine. The percentage of Cox-2 positive cells (chondrocytes in cartilage/fibroblasts, fibrocytes in subchondral bone, and hematopoietic elements in bone marrow) was separately assessed in three joint regions: cartilage, subchondral bone, and subchondral bone marrow.

Statistical analysis

The statistical analysis of data was performed with GraphPad Prism 7.01 (GraphPAd Software Inc., La Jolla, CA, USA) and STATISTICA 5.0 (Statsoft, Tulsa, OK, USA). The distribution of histological scores was analysed by use of the Lilliefors and the Shapiro–Wilk test and showed abnormality without exception; therefore, we performed non-parametric tests in interpretation: the Kruskal–Wallis ANOVA test for multiple and the Mann–Whitney U test (two-tailed) for between-group comparison, the Spearman rank correlation for correlation analysis and the Wilcoxon matched pairs test for matched data series (logarithmically transformed cytokine levels). The threshold of significance was set to p = 0.05, and for each test we also calculated the effect size (z, r in the Mann–Whitney U test, r in Spearman correlation). We applied Holm’s sequential Bonferroni adjustment to the p values obtained in multiple comparisons of the osteoarthritic groups (GrpP, GrpM Lo, GrpM Hi).

Results

Pharmacokinetic study of meloxicam

In the pharmacokinetic pre-study of meloxicam, the following characteristics were recorded: Cmax = 4.79 ± 0.53 μg/mL, tmax =8 ± 3.1 h, t1∕2 = 15.38 ± 2.91 h, MRT = 25.61 ± 2.92 h, AUCt→∞ < 13.5 ± 4.1%.

Characteristic joint lesions of the groups

There were minimal detectable lesions in specimens of GrpCo. In GrpP, tissue lesions were the most severe, with the following key features: erosion, superficial delamination, excavation with matrix loss extending to the mid zone, and denudation with a variable mass of fibrocartilaginous tissue, microfractures of the bone plate. Deformed fibrocartilaginous articular surface was present in one of 11 animals, and severe fragmentation of the calcified cartilage was present in two of 12. Significant (up to 34) fibroblastic transformation of the subjacent bone marrow was present in 5 animals (Figs. 2A–2C).

Figure 2 Histological findings in the studied groups.

(A–C) MIA-induced joint tissue lesions in Grp P. (A) H&E stain, 10x: deformation of the articular surface, cartilage denudation with microfractures (arrow), bone remodeling, and mesenchymal transformation of the bone marrow (arrowhead). (B) H&E stain, 20x: elongated and flattened chondrocytes and extensive zones lacking viable cells (arrow). (C) PAS stain, 10x: superficial zone of cartilage with loss of matrix in the upper one-third (arrow). (D–F) Cartilage and subchondral bone lesions in GrpM Lo. (D) H&E stain, 20x: intact superficial zone with edema and deep fibrillation, disorientation and flattening of the chondrocytes. (E) H&E stain, 10x: erosion with cartilage matrix loss, branched fissure (arrow). (F) H&E stain, 10x: cartilage erosion; vertical, branched fissures (arrow), cysts (arrowhead) and mesenchymal changes affecting up to three-fourths of the bone marrow volume. (G–I) Cartilage and subchondral bone lesions in GrpM Hi. (G) PAS stain, 10x: intact superficial zone, edema, focal matrix condensation (arrow). (H) H&E stain, 10x: disorientation of chondron columns, with cell death, cell clustering and hypertrophy. Bone marrow mesenchymal changes involving approximately one-fourth of the total volume and increased thickening of the subchondral bone marrow. (I) H&E stain, 10x: intact surface with cell death and hypertrophy in the superficial zone, matrix edema, and no marrow changes in the subchondral bone.

In comparison with these results, at GrpM Lo, the tissue lesions were reduced: half of the cases (6/12) had moderate matrix loss with focal rarefaction and condensation of collagen fibers, mild-to-moderate grade of chondrocyte loss and superficial fissures. 4/12 cases presented deep fissures, a more intense matrix loss and collagen fiber condensation, and at 2/12 had denudation with complete erosion. Marked fragmentation of the calcified cartilage could be identified in 2/12 subjects, while important fibroblastic transformation of the bone marrow was seen also in 2/12 specimens (Figs. 2D–2F).

In GrpM Hi, even lesser pronounced histological deteriorations were observed. At 6/11 animals the most prominent changes were superficial matrix discontinuity with a low-grade chondrocyte death or hypertrophy, 4/12 animals showed variable vertical fissures, and only 1/12 presented erosion, important subchondral fragmentation, and fibroblastic transformation (Figs. 2G–2I).

The OARSI histopathology initiative scores

All three osteoarthritic groups had significantly higher CDS, TCDW, SBD and SR scores than did the GrpCo (data shown in Fig. 3). Multiple comparisons by the use of Kruskal–Wallis ANOVA of GrpP, GrpM Lo and GrpM Hi showed a significant effect on CDS (p = 0.003), TCDW (p = 0.005), SBD (p = 0.041). In GrpP, CDS was the highest possible (15) in four of 11 animals. In GrpM Lo, the range of the CDS was 3–12; in GrpM Hi, it was 5–11.

Figure 3 Scatter plot representation of the histological scores.

(A) CDS, cartilage degeneration score; (B) TCDW, total cartilage degeneration width; (C) SBD, calcified cartilage and subchondral bone damage score; (D) SR, synovial reaction. Values shown as median and interquartile range. Significant differences marked with * for p < 0.05, **p < 0.01, *** for p < 0.001. Comparisons for GrpCo shown in boxes, in order for GrpP, GrpM Lo, GrpM Hi.

The histological scores of the study groups, are shown on Fig. 3 and in the Table S1. CDS was significantly lower in GrpM Lo and GrpM Hi than in GrpP: 8.75 (6.5–10.5) and 8 (6–10.5) vs. 12 (11–15), p = 0.012 (z = 2.67, r = 0.56) and p = 0.003, (z = 3.08, r = 0.66). Similarly, the TCDW was significantly lower in GrpM Lo and GrpM Hi than in GrpP: 875 (600–1400) μm and 950 (800–1500) μm vs. 1,600 (1280–2000) μm, p = 0.014 (z = 2.62, r = 0.55) and p = 0.006 (z = 2.99, r = 0.64). The SBD score was significantly lower in GrpM Hi than in GrpP—1 (0–2) vs. 2.5 (2–3), p = 0.011, but the difference between GrpM Lo and GrpP was not significant—1(0.75–2.5) vs. 2.5 (2–3), p = 0.10 (z = 1.63, r = 0.34). CDS, TCDW and SBD scores of GrpM Lo and GrpM Hi were not significantly different. Synovial reaction (SR) scores of the groups were similar (Fig. 3 and Table S1).

Serum cytokines

The serum cytokine values of the various groups are given in Table 1. The between-groups baseline (end of week 3), treatment end-point (end of week 7), and study end-point (end of week 11) comparisons showed that the log-transformed serum TNF α rose in GrpP from week 3 to week 7 (p = 0.049) and IL-6 in GrpM Lo fell from week 3 to week 11 (p = 0.044). All the rest of comparisons gave no significant differences (Table 1).

Table 1 The serum cytokine values in the study groups.

Values expressed as mean ± SE (standard error).

		GrpCo (n = 10)	GrpP (n = 11)	GrpM Lo (n = 12)	GrpM Hi (n = 11)	
IL-6 (pg/mL)	w3	30.4 ± 2.3	26.6 ± 1.6	32.9 ± 3.1*	25.7 ± 1.4	
	w7	32.5 ± 3.3	30.0 ± 1.9	25.4 ± 3.1	26.4 ± 1.7	
	w11	29.2 ± 2.4	28.4 ± 1.9	22.8 ± 2.2*	22.9 ± 3.4	
IL-10 (pg/mL)	w3	7.0 ± 1.7	5.6 ± 1.0	7.9 ± 3.4	3.7 ± 0.9	
	w7	6.6 ± 1.4	7.4 ± 1.7	3.6 ± 0.8	5.1 ± 1.3	
	w11	8.3 ± 2.0	6.3 ± 1.9	3.7 ± 0.7	3.1 ± 0.8	
TNF (pg/mL)	w3	4.0 ± 0.5	3.1 ± 0.8†	3.7 ± 0.7	3.5 ± 0.7	
	w7	3.8 ± 0.5	3.8 ± 0.4†	3.1 ± 0.6	3.2 ± 0.6	
	w11	3.4 ± 0.4	2.8 ± 0.3	2.9 ± 0.3	3.0 ± 0.3	
Notes.

w3, end of week 3 (baseline); w7, end of week 7; w11, end of week 11.

* p < 0.05, shown for paired comparisons of log-transformed IL-6 values in GrpM Lo.

† p < 0.05, shown for paired comparisons of log-transformed TNF values in GrpP.

Cox-2 expression

The Cox-2 immunostaining values for the study groups are given in Table 2. Cox-2 expression was undetectable in GrpCo and was very faint in the subchondral bone sub-region of animals in GrpM Hi. In GrpP, low-grade cartilage staining, and a variable, but higher-grade, subchondral bone and bone marrow expression, were observed. Animals from GrpM Lo had no cartilage reactions and only faint Cox-2 staining in subchondral bone and bone marrow (Fig. 4). The Kruskal–Wallis ANOVA test revealed significant differences between GrpP, GrpM Lo and GrpM Hi both for subchondral bone (p < 0.001) and bone marrow (p < 0.001). Holm-Bonferroni corrected statistics resulted in significantly lower Cox-2 percentages in the subchondral bone in GrpM Hi (p = 0.006, z = 3.08, r = 0.65) and GrpM Lo (p = 0.030, z = 2.46, r = 0.51) than in GrpP. Cox-2 scores of bone marrow were different in GrpM Hi vs. GrpP (p = 0.006, z = 3.12, r = 0.66), and GrpM Lo vs. GrpP (p = 0.016, z = 2.67, r = 0.56) (Table 2).

Figure 4 Cox-2 immunohistochemistry staining of the joint tissues.

(A) Animal from Grp P: bone marrow with positive staining, showing focal cytoplasmic and membrane reaction in mononuclear cells (arrow). (B) Animal from GrpM Lo: cartilage-bone transitional zone with a few Cox-2 positive fibroblasts, fibrocytes, and mononuclear cells (arrows). (C) Animal from GrpM Hi: hypercellular bone marrow, with rare Cox-2-positive cells (arrows).

Table 2 Cox-2 immunostaining scores of the studied groups.

Cox-2 staining expressed as percentage values, expressed as mean ± SE (standard error). Holm-Bonferroni adjusted *p < 0.05, ††p < 0.01 shown for paired comparisons between Grp P and GrpM Lo (marked with*), and GrpP and GrpM Hi (marked with †), respectively. Significance values for comparisons of GrpCo are not shown.

Cox-2 staining score	GrpCo (n = 10)	GrpP (n = 11)	GrpM Lo (n = 12)	GrpM Hi (n = 11)	
Cartilage	0	1.82 ± 0.66	0	0	
Subchondral bone	0	6.54 ± 1.60*,††	1.83 ± 0.58*	0.81 ± 0.55††	
Bone marrow	0	12.18 ± 3.74*,††	1.58 ± 0.58*	0.73 ± 0.48††	

Discussion

Different MIA doses have been used previously to provoke chemically osteoarthritis in rats (Guzman et al., 2003; Dumond et al., 2004; Boudenot et al., 2014; Guingamp et al., 1997; Miyamoto et al., 2016). However, it is known that beside a rapid and dose-dependent decrease of locomotor activity, a secondary, progressive, long-term loss of spontaneous mobility happens only if high doses of MIA (0.3–3 mg) are applied (Guingamp et al., 1997). Considering this finding, we focused our study on the high-dose MIA (4 mg) induced osteoarthritis.

Data accumulated previously suggest that pharmacokinetics of meloxicam in rats is similar to those in humans, the rat being the preferred species for data extrapolation to humans (Busch et al., 1998). However, there is remarkable pharmacokinetic variability in the results among authors: t1∕2 = 13 h in male and t1∕2 = 37 h in female rats (Busch et al., 1998); t1∕2 = 9 h (Aguilar-Mariscal et al., 2007); and t1∕2 = 19–23 h (Aghazadeh-Habashi & Jamali, 2008). Our mean value of Cmax obtained (4.79 μg/mL) was higher than the values reported by other authors: 1.1 μg/mL (after a unique dose of 1 mg/kg (Ochi et al., 2013) or 3.5 μg/mL (after a dose of 0.9 mg/kg) (Aghazadeh-Habashi & Jamali, 2008). The t1∕2 calculated for non-compartmental pharmacokinetics was similar with those obtained by other groups (Busch et al., 1998; Aghazadeh-Habashi & Jamali, 2008). The explanation for the high Cmax might be our formulation: meloxicam was dissolved in (50:50) PEG400:H2O, instead of powder, tablets or aqueous suspension. Considering the long t1∕2 obtained, we concluded that our meloxicam formulation can be administered once daily, and that the steady-state concentration is probably achieved in about five days. The dosage of meloxicam was defined taking into account the human equivalent dose (HED), and the repeated dose toxicity “No Effect Level” (NOEL).

According to Sengupta, a day of a laboratory rat equals 34.8 days of a human, and such a period of 11 weeks post-administration of MIA equals by extrapolation approximately seven years in humans (Sengupta, 2013). Moreover, since the first inflammatory signs in the MIA model appear at the end of the first week, week 11 can be considered the chronic, late-phase of the MIA-triggered inflammatory response. We hypothesized that low-grade inflammation induced by MIA persists in the subchondral bone and bone marrow in the late-phase reaction of osteoarthritis, thus sustaining cartilage destruction. We also proposed that meloxicam could ameliorate not only the cartilage deterioration, but also the lesions provoked on subchondral bone. To test these hypotheses, we used the OARSI-recommended histological approach in evaluating the osteoarthritis and response to meloxicam treatment.

Our major findings were that meloxicam at low-dose (0.2 mg/kg) and high dose (1 mg/kg) had a chondroprotective effect, and the high dose also protected against subchondral bone lesions. The late-phase inflammatory process was alleviated by meloxicam treatment.

The presence of a low-grade, local inflammation was highlighted through Cox-2 expression that was rarely detectable at the level of cartilage, but was more intense in subchondral bone and bone marrow 11 weeks after the MIA-trigger in the GrpP animals. This low-grade inflammation could be an important clue of the progressive degenerative pathways. Cox-2 expression in subchondral bone and bone marrow was significantly suppressed in GrpM Lo (low-dose meloxicam) and virtually absent in GrpM Hi (high-dose meloxicam) animals, which is evidence of the protection of meloxicam in this model.

Concerning the histological assessment, we used standardized approaches because they offer a more precise evaluation of tissue lesions in osteoarthritis (Pritzker et al., 2006; Gerwin et al., 2010).

We chose four scores of the OARSI histopathology initiative that are relevant not only for the depth and width of cartilage degradation, but also for evaluation of the interacting joint regions: the subchondral bone and the synovial membrane. In our study, the Cartilage Degeneration Score (CDS) and the Total Cartilage Degeneration Width (TCDW) were significantly improved in both MXC-treated rat groups. Both the low and high dose of meloxicam were effective in improving the depth (cartilage degeneration score) and extension (total cartilage degeneration width) of the lesions in the MIA-triggered osteoarthritis. In addition, it is even more important that high-dose meloxicam conferred protection also for the deep cartilage and subchondral bone, since this functional unit might be essential in down-regulation of the persistent inflammation and the long-term degeneration pathway.

Application of the subchondral bone damage score in histological assessments may be important in future studies since it is the only simultaneous measure of the deep cartilage-subchondral bone-bone marrow alterations.

Functional models of the joint reveal a complex interplay between cartilage, subchondral bone, and bone marrow. This relationship suggests the possibility of simultaneous investigation of potentially protective drugs both at the levels of cartilage and the subchondral bone (Yuan et al., 2014; Funck-Brentano & Cohen-Solal, 2011). In patients of the Multicenter Osteoarthritis study, bone-marrow changes in osteoarthritis were tightly associated with subchondral bone attrition (Roemer et al., 2010). The time-sequence of histological lesions generated by MIA until day 56, post-administration has been documented (Guzman et al., 2003). Cartilage degradation, fibrillation and chondrocyte degeneration are early phenomena, followed by the involvement of subchondral bone that becomes obvious at the end of the first week after the injection and lasts for at least 56 days (Guzman et al., 2003; Morenko et al., 2004; Pitcher, Sousa-Valente & Malcangio, 2016). The most remarkable consequences seen are the intensification of bone remodelling and the appearance of microfractures and osteophytes, but mesenchymal transformation of bone marrow spaces subjacent to the most affected cartilage zones also is present. MIA also causes a reduction of bone mineral density in the proximal tibia (Boudenot et al., 2014). Histologically, many features of the late-phase lesions of MIA-induced osteoarthritis resemble those of the advanced-phase human disease (Lorenz & Richter, 2006).

Meloxicam has a high anti-inflammatory potential because of its preferential Cox-2 inhibition, but its chondroprotective effect is not unanimously supported by literature reports. Early studies showed that meloxicam is chondroneutral and does not influence proteoglycan synthesis (Engelhardt, 1996). However, in cell-culture studies, meloxicam generated favourable effects on overall proteoglycan synthesis (Blot et al., 2000). In an animal study (Jones et al., 2010), 3 mg/kg meloxicam in monotherapy did not improve cartilage lesions, but it correlated with a higher bone-volume percentage in female Sprague-Dawley rats that underwent knee triad injury. In contrast with these findings, Wen et al. (2013) recently elucidated in anterior crucial ligament trans-section-induced osteoarthritis in Wistar rats a significant improvement of the OARSI and synovial scores with doses of 0.25 mg/kg and 1 mg/kg meloxicam (Wen et al., 2013).

Other Cox-2 selective inhibitors, such as celecoxib, improved proteoglycan synthesis and turnover and synovial release of IL-1β and TNFα in samples from human patients who underwent knee-replacement surgery (De Boer et al., 2009). Celecoxib in combination with rebamipide is highly efficient in reducing osteoclast number in the subchondral bone marrow in MIA-induced rat osteoarthritis (Moon et al., 2013).

The mechanisms of disease progression in the MIA-induced osteoarthritis are not well-known; however, the time-specific participation of pro-inflammatory molecules in cartilage-bone crosstalk has been documented (Funck-Brentano & Cohen-Solal, 2011). Possibly, the first inflammatory burst after MIA injection, and then a continuing low-grade inflammation nested in the subchondral bone and bone marrow, contribute to the catabolic reprogramming of chondrocytes and to the emergence of inflammatory transcriptional “go-signals” (Liu-Bryan & Terkeltaub, 2015). This phenomenon could be an explanation for the overall improvement of cartilage and subchondral bone-damage by the high-dose meloxicam in our experiments. Bone marrow stimulation followed by implantation of acellular biomaterials significantly improved cartilage repair in several preclinical studies (Pot et al., 2016).

Much evidence supports the pathogenic role of pro-inflammatory (IL-1, IL-6, TNFα, IL-15, IL-17) and anti-inflammatory (IL-4, IL-10) cytokines in osteoarthritis (Kapoor et al., 2011; Wojdasiewicz, Poniatowski & Szukiewicz, 2014); however, the presence of a systemic inflammation in osteoarthritis has not been confirmed unanimously. In the cartilage, TNFα causes a focal tissue loss due to the fact that chondrocytes wear variable amounts of p75 TNFα receptor (Westacott et al., 2000). Local chemokines, such as CCL20 can induce IL-6 and Cox-2 in explanted donor and osteoarthritic chondrocytes (Alaaeddine et al., 2015). Increased circulating levels of IL-6 and IL-10 have been reported in painful osteoarthritis (Imamura et al., 2015), but this reaction might be absent in the late-disease phase. Isolated mechanical injuries of the joint may lead to self-perpetuating, chronic local inflammatory reactions (Sokolove & Lepus, 2013). In our placebo-group, the levels of serum TNFα ascended from week 3 to week 7. The source of this systemic elevation might have been the progressive cartilage destruction and erosion. Meloxicam did not influence the IL-10 and the TNFα levels neither in GrpM Lo nor in GrpM Hi. In parallel, IL-6 at GrpM Lo fell about 40% from week 3 to week 11, but this effect was not seen with GrpM Hi. The decrease of IL-6 could bear a therapeutic benefit in GrpM Lo; however, it was not characteristic of the high-dose meloxicam with a more pronounced histological effect. A significant drawback for interpretation of these data is the source of cytokines used in our study: the synovial fluid instead of serum probably would be more useful in establishing the dose-related local effects. However, comparison of cytokine profiles of the synovial fluid and the cartilage proved difference (Tsuchida et al., 2014).

There are different opinions in the literature regarding the applicability of MIA-models to study of the spontaneous progression of osteoarthritis. According to some authors, the consequences of MIA injection imitate an inflammatory arthritis rather than spontaneous disease (Teeple et al., 2013). The transcriptional profile overlap between human and MIA-induced rat osteoarthritis is poor, but the comparison was made only for cartilage (Teeple et al., 2013); others studies, in contrast, emphasize that the MIA-model is minimally invasive, rapid, and reproducible, with comparable degenerative and histological changes to the human anterior cruciate ligament-induced osteoarthritis (Thysen, Luyten & Lories, 2015; Naveen et al., 2014).

Our data bring evidence that both low dose (0.2 mg/kg) and high dose (1 mg/kg) meloxicam are efficient alternatives, with reference both to the depth (CDS, SBD) and extension (TCDW) of the lesions in MIA-triggered osteoarthritis. In addition, it is even more important that high-dose meloxicam conferred protection also for the deep cartilage and subchondral bone, since this functional unit might be essential in down-regulation of the persistent inflammation and the long-term degeneration pathway.

Conclusions

In this rat model of MIA-induced late-phase osteoarthritis, both low-dose and high-dose meloxicam had a chondroprotective effect, and the high dose also protected against subchondral bone lesions. The importance of this finding lies in the assumption that subchondral bone and bone marrow alterations sustain and perpetuate the deterioration of cartilage. Although findings in the rat model cannot be directly extrapolated to human disease, the favourable changes seen at the level of subchondral bone and bone marrow suggest that meloxicam therapy, especially in high doses, can attenuate chronic-phase disease progression.

More evidence, including immunohistochemical characterization of the inflammatory infiltrate and fibroblastic transformation, is needed to highlight the specific action of meloxicam on subchondral bone and bone marrow.

Supplemental Information

Supplemental Information 1 Scatter plot representation of the histological scores

(A) CDS, cartilage degeneration score; (B) TCDW, total cartilage degeneration width; (C) SBD, calcified cartilage and subchondral bone damage score; (D) SR, synovial reaction. Values shown as median and interquartile range. Significant differences marked with * for p < 0.05, ** p < 0.01. Comparisons for GrpCo shown in boxes, in order for GrpP, GrpM Lo, GrpM Hi.

Click here for additional data file.

Table S1 Histological scores of the studied groups

Values expressed as mean with median, minimum– maximum values in brackets. Holm-Bonferroni adjusted * p < 0.05, **, ††p < 0.01 for paired comparisons between groups II and III (marked with *), and groups II and IV (marked with †), respectively. TCDW given in μm. Significances for comparisons of Group I are not shown. Scores are those of the OARSI histopathology initiative: CDS, cartilage degeneration score; TCDW, total cartilage degeneration width; SBD, calcified cartilage and subchondral bone damage score; SR, synovial reaction.

Click here for additional data file.

Data S1 Raw data table

Click here for additional data file.

We thank Professor Daniela-Lucia Muntean for the organizational support, Dr. Gabriela Marcus, Ana Popeiu and Teodora Popeiu for their participation in animal treatment and surveillance, and Dr. Miklós Bob for the kind interpretation of radiological data.

Additional Information and Declarations

Competing Interests

Author Contributions

Animal Ethics

Data Availability

The authors declare there are no competing interests.

Előd Nagy conceived and designed the experiments, analyzed the data, contributed reagents/materials/analysis tools, wrote the paper, prepared figures and/or tables.

Enikő Vajda conceived and designed the experiments, performed the experiments, analyzed the data, contributed reagents/materials/analysis tools, prepared figures and/or tables.

Camil Vari conceived and designed the experiments, reviewed drafts of the paper.

Sándor Sipka reviewed drafts of the paper.

Ana-Maria Fárr performed the experiments.

Emőke Horváth performed the experiments, contributed reagents/materials/analysis tools, wrote the paper, prepared figures and/or tables.

The following information was supplied relating to ethical approvals (i.e., approving body and any reference numbers):

The study obtained the approval of the Ethics Committee of the University of Medicine and Pharmacy Targu-Mures (no.132/23.12.2013).

The following information was supplied regarding data availability:

The raw data has been supplied as a Supplementary File.

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
