# Peer review of "Meloxicam ameliorates the cartilage and subchondral bone deterioration in monoiodoacetate-induced rat osteoarthritis"

_PeerJ, doi:10.7717/peerj.3185_

## Round 0.1 · original submission · Major Revisions

· Academic Editor

Major Revisions

Your manuscript has been reviewed by two experts in the field. As you can see, they have raised many points to be corrected. Please, do the corrections and note that your manuscript will be again revised and to accelerate the second review I strongly suggest you to indicate in the rebuttal letter the location of all the changes made in the original version.

Please also be sure to address the concerns of reviewer 1 relating to the euthanasia method and blood sampling.

·

Basic reporting

The manuscript is presenting drawbacks, which should be taken into consideration before publication in PeerJ.

This preclinical in vivo study aimed originally to evaluate the curative medication effect of a low or high dose of meloxicam on cartilage and subchondral bone deteriorations in a rodent model of chemically (MIA)-induced osteoarthritis. The aim and the results are interesting but the study is presenting several Basic Reporting failures, namely:

- Objectives and hypothesis are well defined but the relevance of doing a pharmacokinetic study is missing.
- A major lack of rigor in the manner to cite references in the text (there is a lot of heterogeneity, particularly in the introduction section), for ex.: (Dumond et al 2004). The; . (Dumond et al, 2014) The ...; (Dumond et al. 2014) or again (Dumond et al., 2014)

*Abstract

- Objective: The abbreviation of versus (vs.) should always be in italics, as this is a Latin (foreign language) citation. This error is repeated throughout the text.

- Methods: Monosodium iodoacetate (MIA) is well known to induce osteoarthritis in stifle of rodent similar to human disease. In this study, the authors often use the term arthritis in combination with osteoarthritis. There is a misunderstood for the authors between these two different pathologies. This mistake is repeated several times in the article. Moreover, the term knee is use to describe a joint in human. In animal, we should use the term stifle. This error is repeated throughout the text.

-Results: It lacks all the statistical P values for the reader.

*Introduction

L52: Please add the abbreviation of osteoarthritis in brackets.

L70: Please define the term cox-2.

L71: Same comment for IL-1β.

L73 : Please replace condyli by condyle.

*Figures

Are not really useful (such as Fig 2 and 4), and of poor quality (such as Fig 3).

*Discussion

L276: The term arthritis is inappropriate.

L282: The term arthritis is inappropriate.

-The authors do not discuss the PK results obtained and the major limitations concerning this part of the study.

-Moreover, meloxicam is well known to produce analgesic effects and pain is omnipresent in osteoarthritis disease. The authors never discuss the possible difference in the degree of pain observed between the four groups of rats during the curative treatment period.

Experimental design

The aim and the results are interesting but the study is presenting several Methodological failures, namely:

- Objectives and hypothesis are well defined but the relevance of doing a pharmacokinetic study is missing.
- Experimental design: The additional 4 weeks after treatment cessation at the end is not supported in the methodology or in the discussion, as well as the dose of mososodium iodoacetate injected in the stifle of the rats.
- Statistics must be reviewed for cytokines quantification.
- Cytokine analysis: Lack of reference for cytokine concentrations from normal rats serum. Injection of saline in control group seems not so neutral, causing inflammation.
- The management of pain is not assayed despite the well-known analgesic effect of the drug used.

- There are major confusion in the number of animals used for the study. We understand from Figure 1 that 2 animals were excluded, but there is no mention for which reason. Without such information, the exclusion could be associate to data cherry picking.

- There is no indication that the treatments administration was blinded.

- Blood sampling volume must be indicated every where, when required. This is particularly important for the PK study, as such repeated blood sampling in rats has major ethical regulations.

- Most of all, the reviewer has major problem with the method of euthanasia used, namely "excessive halothane inhalation". In Canada, the CCAC does no more accept such method of inhaled anesthetics only as humane method of euthanasia, the easons being the stress and the aversion caused to gas exposure, the length before getting euthanasia, and the technical difficulties related to get efficient overdose in a safe environment. As stated by the CCAC "In general, overdose of an inhalation anesthetic agent is an effective method of euthanasia for many species. However, time to death is quite lengthy, and therefore use of a second procedure to ensure death of the animal is recommended once the animal is unconscious as a result of the anesthetic", and, in my appreciation, the authors must detail the duration to gas exposure, with concentration of exposure as well as signs assessed by the veterinarian for confirming death.

Methodology

Figure 1: The benefit to extend the study 4 additional weeks at the end without treatment is not supported.

L113: A comma is underlined.

L119-128: The osteoarthritis model: The choice to inject 4 mg of MIA in a volume of 50 uL of saline when the reference cited in the introduction (Guzman et al., 2003) used a dose of 2 mg in 50 uL of saline is not justified. Same thing with the other cited reference in the introduction (Dumond et al., 2004) which injected a dose of 0.03 mg of the chemical agent. The references in the introduction are not relevant with the experimental design. Moreover, no references are cited in this part of the methodology supporting MIA dose.

L135-139: Meloxicam doses (low and high) have been established on which references? A pilot study have been done before this study in the aim to determine the doses of meloxicam?

L141-152: Pharmacokinetic (PK) study of meloxicam: The goal and objective of this part of study are missing. This PK study is not supported before in the introduction and not discussed later. How the timepoints for blood collection have been defined? Are they optimal for a PK study using meloxicam in male rats? Why only the high dose have been modeling? Which PK model have you used to determine the parameters of meloxicam? The results shown in L153-154 concerning the PK study should not be found in this part of the article.

L196-206: Statistical analysis: For your personal information, you don’t need to verify the normal distribution for ordinal values like as histological score. Have you tried to manipulate cytokine data (for example in semi-logarithmic, logarithmic or exponential transformation) in order to get a normal distribution?

Validity of the findings

A major critic is related to the semi-quantitative score for determining COX-2 reactivity:

A semi-quantitative scoring system was applied: score 0, no reaction; score 1, faint reaction, 1-5% of cells positive; score 2, 6-25% of the cells positive; score 3, 26-50% of the cells positive; score 4, >50% of the cells (chondrocytes in cartilage/fibroblasts, fibrocytes in subchondral bone, and hematopoietic elements in bone marrow) positive.

As the range of values presented in Table 3 is from 0 to 3, such classification induces a potential bias toward the Placebo group which would have presented 6% of positive cells to get a score of 2, whereas the Meloxicam groups would get 5% and getting a score of 1. It would have been a better approach to compare the percentage of positive cells between groups.

Results
L234: Inter-groups significant difference?

Table 1: The fail to demonstrate some difference in cytokine concentrations between the four groups seem to be related with the fact that the injection of saline appears to be not so neutral, causing some inflammation. What is the normal range of concentration for each cytokine in serum from naive rats? Or it could be related to a variable timing of expression in cytokines. Do you think that the choice of serum is the appropriate biological fluid to quantify cytokines in this study? All these limitations need to be discussed.

L257-260: The description of the statistical test used in the results section is inappropriate.

Reviewer 2 ·

Basic reporting

Basic reporting
Clear, unambiguous, professional English is used throughout the manuscript.
Line 53 -57: Suggests that the role of subchondral bone in the pathogenesis of OA is a recent finding, however it has been argued for decades that subchondral bone may drive disease onset and/ or progression. In particular, increased cytokine production by osteoblasts in OA was reported as early as 1997 and 1998 by Westacott and Hilal, respectively. This should be reflected in the text and references, along with the more recent discoveries by Berenbaum, Yu and Yuan.
Line 61-68: The authors describe the histological changes of MIA-induced OA by Guzman, however it would be useful to include more information on the timing of disease progression, so that the reader understands why treatments were administered at the time points chosen in this study. It is explained in the discussion but this needs to be made clearer in the introduction, along with a clearer explanation of study design in general.
Introduction shows context, but the aim of the study is not clearly stated. Furthermore, argumentation concerning the reason that this model represents chronic OA and the used dosage Meloxicam is lacking. The paragraph which describes the histological grading systems (88-95) is too extensive and outside the context.
The structure conforms with the PeerJ standard.
Figures are relevant and high quality, however some changes are needed (see below).
Figure 2 legend needs to be simplified. The descriptions of histological findings are long and repeated in the results section. A scale bar and uniform orientation of images (with cartilage surface at the top) would aid interpretation of results by the reader.
Figure 4: Re-order images so that they are in keeping with Table 2. As with figure 2, include a scale bar and same orientation.
Raw data is supplied.

Experimental design

Experimental design
Original primary research within scope of the journal.
The research question is not well defined. The comparison between low- and high-dosage Meloxicam is mentioned in the abstract, but the research question is lacking in the introduction.
It could be more emphasized how this research fills an identified knowledge gap.
Rigorous investigation performed to a high technical and ethical standard.
Methods described with sufficient detail and information to replicate in general. Exceptions are: -The sample size calculation: it is not clear which means are used.
-The presence of OA confirmed by radiography: the grading system used needs to be stated in the methods section along with the introduction.
A key finding of this paper is the reduction in subchondral bone lesions with high-dose meloxicam treatment, and that subchondral bone expression of Cox-2 is shown to be reduced with both high and low-dose meloxicam, compared with the placebo. The authors suggest in the discussion that local low-grade inflammation was highlighted through Cox-2 expression and subsequently suppressed by meloxicam- which is a Cox-2 specific inhibitor. More evidence is needed to demonstrate subchondral bone inflammation and its suppression by meloxicam, such as histological scoring of inflammatory features (infiltrating cells) and/ or immunohistochemical staining for inflammatory cytokines and CD markers.

Validity of the findings

Validity of the findings
Data is robust, statistically sound and controlled.
The results in the paragraph named “characteristic joint lesions of the groups’ is hard to follow. The reader should be directed to the Figure panel at the beginning of the paragraph so that descriptions and comparisons between example images can be followed.
The discussion seemed to be an overview about different aspects of e.g. (MIA-induced) OA models, Meloxicam, and systemic inflammation, each of them represented in a separate paragraph. There is no logical flow and what is stated is not well connected to the findings of this paper.

Additional comments

Additional comments:
Abstract
Objective: The comparison in the results is between low-dose and placebo and between high dose and placebo. So Low-dose AND high-dose meloxicam treatment instead of versus.
Conclusion: Is 11 weeks already chronic?
Main text
Line 70-72: You mention two times increased Cox-2 activity.
Line 87: Where is the aim of the study?
Line 111: Means of which outcome?
Line 131: Which grading system or guidelines?
Line 136: Are these dosages based on previous research?
Line 243-244: This is the aim of the study?
Line 247: Table 1 not Table 2.
Line 267-271: Needs to be including in the introduction as well as discussion.
Line 280: Explain "late phase" inflammation in introduction
Line 291: Together with the next paragraph?
Line 294,295 and line 298,299,300 are basically the same
Line 318: What is the connection of this paragraph with this paper?
Line 322, 332, 337: Relevance to data presented?
Line 346-349 Is not of additive value
Line 355: Include in paragraph about cytokine findings.
Line 356: Include in paragraph 338-349
Line 365: Alternative for what?
Line 364: Already stated in first part of discussion.
Figures and tables
Figure 2: Too long. So d-f = low dose and g-i is high dose?
Figure 3: What is the purpose of the boxes with asterisks?
Table 1: Why not present data as mean and standard deviation?

---

## Round 0.2 · accepted · Accept

· Academic Editor

Accept

Thank you for revising the manuscript. I have looked at the rebuttal letters and the revised manuscript. I realize that you have replied to all the comments, suggestions and corrections. Please, only check the number of rats (36 and 34). I realize that 2 rats missing in the experiment were the ones that you cite in the rebuttal letter (they were excluded from the experiment because they did not present sufficient the inflammatory response.